Back from the dead; the curious tale of the predatory cyanobacterium Vampirovibrio chlorellavorus

Soo Rochelle M. 1
Woodcroft Ben J. 1
Parks Donovan H. 1
Tyson Gene W. 1 2
Hugenholtz Philip 1 3 p.hugenholtz@uq.edu.au
1 Australian Centre for Ecogenomics, School of Chemistry and Molecular Biosciences, The University of Queensland , St Lucia, QLD , Australia
2 Advanced Water Management Centre, The University of Queensland , St Lucia, QLD , Australia
3 Institute for Molecular Bioscience, The University of Queensland , St Lucia, QLD , Australia
Chistoserdova Ludmila
Electronic publication date: 2015 May 21
Publication date: 2015
Volume: 3
Electronic Location ID: e968
Received 2015 Mar 26; Accepted 2015 Apr 30
Copyright: © 2015 Soo et al.
Copyright year: 2015
Copyright holder: Soo et al.
License: This is an open access article distributed under the terms of the Creative Commons Attribution License, which permits unrestricted use, distribution, reproduction and adaptation in any medium and for any purpose provided that it is properly attributed. For attribution, the original author(s), title, publication source (PeerJ) and either DOI or URL of the article must be cited.
License URL: https://creativecommons.org/licenses/by/4.0/

Keywords: Cyanobacteria, Melainabacteria, Predatory bacteria, Vampirovibrio chlorellavorus, Chlorella vulgaris, Obligate predator, Epibiotic

Funding: Australian Research Council ARC-DP120103498 Australian Centre for Ecogenomics ARC Queen Elizabeth II fellowship ARC-DP1093175 Australian Postgraduate Award Natural Sciences and Engineering Research Council of Canada The project was supported by the Australian Research Council (ARC) through project ARC-DP120103498, strategic funds from the Australian Centre for Ecogenomics; G.W.T. is supported by an ARC Queen Elizabeth II fellowship [ARC-DP1093175]; R.M.S is supported by an Australian Postgraduate Award (APA); D.H.P. is supported by the Natural Sciences and Engineering Research Council of Canada. The funders had no role in study design, data collection and analysis, decision to publish, or preparation of the manuscript.

==============================
An uncultured non-photosynthetic basal lineage of the Cyanobacteria, the Melainabacteria, was recently characterised by metagenomic analyses of aphotic environmental samples. However, a predatory bacterium, Vampirovibrio chlorellavorus, originally described in 1972 appears to be the first cultured representative of the Melainabacteria based on a 16S rRNA sequence recovered from a lyophilised co-culture of the organism. Here, we sequenced the genome of V. chlorellavorus directly from 36 year-old lyophilised material that could not be resuscitated confirming its identity as a member of the Melainabacteria. We identified attributes in the genome that likely allow V. chlorellavorus to function as an obligate predator of the microalga Chlorella vulgaris, and predict that it is the first described predator to use an Agrobacterium tumefaciens-like conjugative type IV secretion system to invade its host. V. chlorellavorus is the first cyanobacterium recognised to have a predatory lifestyle and further supports the assertion that Melainabacteria are non-photosynthetic.

Introduction

Predatory microorganisms attack and digest their prey, which can be either bacteria or microbial eukaryotes (Coder & Starr, 1978; Stolp & Starr, 1963). They have been found in a range of environments, including terrestrial, freshwater, estuaries, oceans, sewages and animal faeces (Jurkevitch, 2007). Microbial predators have been classified as obligate (unable to grow in the absence of prey) or facultative (able to grow as a pure culture without the presence of prey). In addition they can be periplasmic (penetrate and attach to the inner membrane), epibiotic (attach to the outside), endobiotic (penetrate the cytoplasm) or wolf-pack (swarming as a ‘wolf-pack’ towards prey, which they kill and degrade) (Pasternak et al., 2013; Velicer, Kroos & Lenski, 2000). To date, four bacterial phyla harbour microbial predators; the Proteobacteria, Actinobacteria, Bacteroidetes and Chloroflexi (Casida, 1983; Kiss et al., 2011; Saw et al., 2012; Stolp & Starr, 1963).

In 1972, Gromov and Mamkaeva first described the predatory nature of Bdellovibrio chlorellavorus towards the microalgae Chlorella vulgaris in a Ukrainian freshwater reservoir (Gromov & Mamkaeva, 1972). They reported that co-inoculation of the alga and bacterium resulted in clumping and colour change of algal cells, formation of refractile bodies and finally algal cell death. However, unlike other Bdellovibrio species that invade the periplasm of Gram-negative bacteria, B. chlorellavorus only attached to the surface of C. vulgaris, producing peripheral vacuoles in the alga followed by a gradual dissolution of the infected cell contents (Coder & Goff, 1986). This distinct mode of predation called into question the classification of B. chlorellavorus as a Bdellovibrio (Coder & Starr, 1978) resulting in its reclassification as Vampirovibrio chlorellavorus in 1980, although its higher level assignment to the Deltaproteobacteria was retained (Gromov & Mamkaeva, 1980).

Co-cultures of V. chlorellavorus and C. vulgaris were deposited in three culture collections in 1978 (Coder & Starr, 1978). However, to the best of our knowledge there are no reports of successful resuscitation of the organism from lyophilised material. The only subsequent studies of V. chlorellavorus were based on co-cultures obtained directly from the investigators who originally enriched the bacterium (Coder & Goff, 1986; Mamkaeva & Rybal’chenko, 1979). The American Type Culture Collection (ATCC) was able to successfully extract DNA from one of the 32 year-old lyophilised co-cultures and sequence the 16S rRNA gene of V. chlorellavorus (Genbank acc. no. HM038000). Comparative analyses of this sequence indicate that V. chlorellavorus is actually a member of the phylum Cyanobacteria rather than the Proteobacteria according to the Greengenes (McDonald et al., 2012) and Silva (Quast et al., 2013) taxonomies. This may explain why the culture could not be revived as Cyanobacteria are notoriously difficult to resuscitate from lyophilised material (Corbett & Parker, 1976). More specifically, V. chlorellavorus is a member of a recently described basal lineage of non-photosynthetic Cyanobacteria, the class Melainabacteria (Soo et al., 2014), originally classified as a sister phylum (Di Rienzi et al., 2013). Here, we report the near-complete genome of V. chlorellavorus sequenced directly from a 36-year-old vial of co-cultured lyophilised cells, confirm its phylogenetic position in the Cyanobacteria, and infer the molecular underpinnings of its predatory life cycle.

Materials and Methods

Sample collection

Co-cultured Vampirovibrio chlorellavorus and Chlorella vulgaris (NCIB 11383) (deposited in 1978 by Coder and Starr) were obtained as lyophilised cells from the National Collection of Industrial, Food and Marine Bacteria (NCIMB), Aberdeen, Scotland.

Genomic DNA extraction

Genomic DNA (gDNA) was extracted from lyophilised cells using a MoBio Soil Extraction kit (MoBio Laboratories, Carlsbad, California, USA). gDNA was quantified using a Qubit 2.0 fluorometer (Life technologies, Carlsbad, California, USA). One ng of the gDNA was used to construct a paired-end library with the Illumina Nextera XT DNA Sample Preparation kit according to protocol but with double size selection to obtain an insert size of 300–800 bp (Quail, Swerdlow & Turner, 2009). The library was sequenced on an Illumina Miseq system using the Miseq Reagent Kit v3 according to manufacturer’s instructions.

Genome assembly, completeness and contamination

Sequencing reads were processed with FastQC to check for quality (http://www.bioinformatics.babraham.ac.uk/projects/fastqc/) and Illumina Nextera adaptors were removed using FASTX-Toolkit (http://hannonlab.cshl.edu/fastx_toolkit/). Reads were parsed through GraftM (https://github.com/geronimp/graftM) version r2439db using the May, 2013 version of the Greengenes database 97% OTUs (operational taxonomic units) as a reference (McDonald et al., 2012) to identify those containing parts of 16S or 18S rRNA genes using default parameters. The 5′ end of all reads was trimmed (∼20bp) to remove low-quality sequence and paired reads were assembled into contigs with a kmer size of 63 using CLC Genomics Workbench v7.0 (CLC bio, Aarhus, Denmark). The statistical package R with ggplot2 (https://github.com/hadley/ggplot2) was used to plot GC content against coverage allowing contigs belonging to the V. chlorellavorus genome to be identified. A discrete cluster of contigs with >180 × coverage and a GC range of 42–54% was identified as belonging to V. chlorellavorus, while contigs with <180 × coverage were assigned to C. vulgaris (Fig. S1). BLASTN (Altschul et al., 1990) (v2.2.29+) using default settings was used to verify that contigs with >180 × coverage had homology to bacterial sequences with NCBI’s non-redundant database. Additionally, the 16S rRNA gene was identified using Prokka v1.8 (Seemann, 2014) and a BLASTN search was used to identify the closest neighbour in the May, 2013 version of the Greengenes database (McDonald et al., 2012). The completeness and contamination of the genome belonging to V. chlorellavorus was examined using CheckM v0.9.5 (Parks et al., 2014a) with a set of 104 conserved bacterial single-copy marker genes (Soo et al., 2014). IslandViewer was used to identify genomic islands (Langille & Brinkman, 2009) with the SIGI-HMM programme (Waack et al., 2006).

Plasmids were identified using the ‘roundup’ mode of FinishM git version 5664703 (https://github.com/wwood/finishm), using raw reads as input, a kmer length of 51bp and a coverage cutoff of 15. A combination of manual inspection of the assembly graph generated using the ‘visualise’ mode and automated assembly with the ‘assemble’ mode confirmed that the contig ends unambiguously joined together (i.e., they joined together and to no other contig ends) and that the two plasmid contigs originally assembled with CLC were otherwise free of mis-assemblies. Plasmids were also confirmed by the annotation of multiple transfer (tra) genes by the Integrated Microbial Genomics Expert Review (IMG/ER) system (see below).

Genome annotation

The V. chlorellavorus genome was submitted to IMG/ER for annotation (Markowitz et al., 2009) and has been deposited at JGI [JGI IMG-ER:2600254900]. The genome was also annotated with prokka v.1.8 (Seemann, 2014) and the Uniref 90 database (Suzek et al., 2007). KEGG maps (Kanehisa et al., 2004) and gene annotations were used to reconstruct the metabolism of the V. chlorellavorus genome. Individual genes that were annotated as ‘hypothetical protein’ or had been potentially misannotated based on the annotation of surrounding genes were further explored through BLASTP searches against the NCBI-nr database. A metabolic cartoon was prepared in Adobe Illustrator CS6.

The methyl-accepting chemotaxis proteins identified by IMG-ER were submitted to InterProScan5 (Jones et al., 2014) to determine chemotaxis protein domains. Putative genes were annotated with the dbCAN web server (Yin et al., 2012) to identify glycoside hydrolases and checked against the IMG annotations and BLAST results. The MEROPS server (Rawlings et al., 2014) was used to identify putative peptidases in V. chlorellavorus using batch BLAST.

A Genbank file for V. chlorellavorus was generated through the xBASE website (Chaudhuri et al., 2008). The ribosomal proteins, chaperones and transcriptional and translational proteins of V. chlorellavorus were used as representatives of recognised highly expressed genes to identify other putatively highly expressed genes in the genome using PHX (predicted highly expressed) analysis using the standard genetic code (http://www.cmbl.uga.edu/software/phxpa.html; Bhaya et al., 2000; Karlin & Mrázek, 2000). Putatively horizontally transferred (alien) genes were identified by their atypical codon usage from the genome average also using PHX analysis.

Genome tree

A bacterial genome tree was inferred in order to establish the phylogenetic relationship of the V. chlorellavorus genome. A set of 5,449 bacterial genomes previously identified as being of exceptional quality were used to establish a set of bacterial marker genes suitable for phylogenetic inference (Parks et al., 2014a). An initial set of 178 single copy genes present exactly once in >90% of the trusted genomes (found in >90% of the genomes) was identified using the Pfam (Finn et al., 2014) and TIGRFAMs (Haft, Selengut & White, 2003) annotations provided by the Integrated Microbial Genomes v.4.510 (IMG; Markowitz et al., 2014). The same protein family may be represented in both Pfam and TIGRFAMs. Families from these two databases were considered redundant if they matched the same genes in >90% of the trusted genomes, in which case preference was given to the TIGRFAMs families. Genes present multiple times within a genome were considered to have congruent phylogenetic histories if all copies of the gene were situated within a single conspecific clade within its gene tree. From the 178 initial genes, 69 were removed from consideration as they exhibited divergent phylogenetic histories in >1% of the trusted genomes (Table S1). The remaining 109 genes were identified across an expanded set of 7,732 bacterial genomes, including all known Melainabacteria genomes along with an outgroup of 169 archaeal genomes using Prodigal v2.60 (Hyatt et al., 2012) to identify all genes and HMMER v3.1b1 (http://hmmer.janelia.org) to assign genes to Pfam and TIGRFAMs families. Gene assignment was performed using model specific cutoff values for both the Pfam (-cut_gc) and TIGRFAMs (-cut_tc) HMMs. For both the individual gene trees and concatenated genome tree, genes were aligned with HMMER v3.1b1 and phylogenetic inference performed with FastTree v2.1.7 (Price, Dehal & Arkin, 2009) under the WAG + GAMMA model. Support values for the bacterial genome tree were determined by applying FastTree to 100 bootstrapped replicates (Felsenstein, 1985). The 16S rRNA gene tree was constructed as previously described (Soo et al., 2014). Briefly, the 16S rRNA gene from V. chlorellavorus was aligned to the standard Greengenes alignment with PyNAST (McDonald et al., 2012). Aligned sequences and a Greengenes reference alignment, version gg_13_5 were imported into ARB and the V. chlorellavorus sequence alignment was corrected using the ARB EDIT tool. Representative taxa (>1,300 nt) were selected for constructing the alignments, which were exported from ARB (Ludwig et al., 2004) with Lane mask filtering. Neighbour joining trees were calculated from the mask alingments with LogDet distance estimation using PAUP*4.0 (Swofford & Sullivan, 2003) with 100 bootstrap replicates. Maximum parsimony trees were calculated using PAUP*4.0 (Swofford & Sullivan, 2003) with 100 bootstrap replicates. Maximum likelihood trees were calculated from the masked alignments using the Generalized Time-Reversible model with Gamma and I options in RAxML version 7.7.8 (Stamatakis, 2006) (raxmlHPC-PTHREADS -f a -k -x 12345 -p 12345 -N 100 -T 4 -m GTRGAMMAI). Bootstrap resampling data (100 replicates) were generated with SEQBOOT in the phylip package (Felsenstein, 1989) and used for 100 bootstrap resamplings. Generated trees were re-imported into ARB for visualisation.

Phylogenetic trees for virB4 and fliI genes

VirB4 sequences were obtained from Guglielmini, de la Cruz & Rocha (2013). The phylip file (figure3_mafft_alignment.phy) obtained from the DRYAD database was converted to an HMM using HMMer v3.1b1 (http://hmmer.janelia.org) and the VirB4 sequences from V. chlorellavorus was aligned to the HMM. The aligned sequences were used to construct a phylogenetic tree with phyml (v3.1) (Guindon et al., 2010) using default settings (Guglielmini, de la Cruz & Rocha, 2013).

The HMM for TIGR03496 (FliI_clade 1) was used to identify fliI genes from 2,256 finished genomes in the IMG database v4 and the 12 Melainabacteria genomes, including V. chlorellavorus. A phylogenetic tree of the fliI genes was constructed using FastTree (version 2.1.7) with default settings (Price, Dehal & Arkin, 2009).

Comparison of V. chlorellavorus to other predatory bacteria

The presence of orthologues for differentiating predatory and non-predatory bacteria as described in Pasternak et al. (2013) were identified in the V. chlorellavorus genome using BLASTP (Altschul et al., 1990) against the OrthoMCL DB v4 (Chen et al., 2006) with an e-value threshold of 1e-5.

Comparison of V. chlorellavorus to other Melainabacteria genomes

Eleven Melainabacteria genomes were compared to the V. chlorellavorus genome (Di Rienzi et al., 2013; Soo et al., 2014). COG profiles were constructed using homology search between putative genes predicted with Prodigal v2.60 (Hyatt et al., 2010) and the 2003 COG database (Tatusov et al., 2003). Genes were assigned to COGs using BLASTP (v2.2.22) with an e-value threshold of 1e-2, an alignment length threshold of 70% and a percent identity threshold of 30%. The relative percentage of a COG category was calculated in relation to the total number of putative genes predicted for each genome. STAMP v2.0.8 (Parks et al., 2014b) was used to explore the resulting COG profiles and create summary plots.

Results and Discussion

Genome summary

A total of 701.2 Mbp of shotgun sequence data (2 × 300 bp paired-end Illumina) was obtained from DNA extracted from a co-culture of Vampirovibrio chlorellavorus and Chlorella vulgaris (NCIB 11384). A search of the unassembled dataset for 16S rRNA sequences revealed 333 reads mapping to V. chlorellavorus (16 chloroplast, 3 mitochondria). No matches to other microorganisms were identified. Sequence reads were assembled into 113 contigs comprising 3.2 Mbp. Ordination of the data by GC content and mapping read depth revealed a high coverage cluster of contigs comprising ∼94% of the data (Fig. S1). These contigs were inferred to belong to V. chlorellavorus by the presence of a 16S rRNA gene on one of the contigs (see below) and low coverage contigs were inferred to belong to the C. vulgaris by best matches to reference Chlorella genomes. Inspection of the assemblies showed no evidence for microheterogeneity (SNPs, indels) in the V. chlorellavorus contigs suggesting that it was a pure bacterial strain. After manual curation, the genome of V. chlorellavorus was represented by 26 contigs comprising a total of 2.91 Mbp with an average GC content of 51.4% and two plasmids comprising ∼72 Kbp and ∼50 Kbp were identified which contained genes for conjugative gene transfer (see below). These plasmids had mapping coverage similar to the genomic contigs suggesting that they are low-copy. The genome was estimated to be near-complete with low contamination according to CheckM (Parks et al., 2014a) suggesting that the fraction of missed genes in contig gaps was minimal. The protein coding density of the genome is 87.1% and predicted to encode 2,847 putative genes, 41 tRNA genes which represent all 20 amino acids and one rRNA operon (only the 16S and 23S rRNA genes were identified). Approximately two thirds (69.9%) of the putative genes can be assigned to a putative function and half (53.2%) can be assigned to a COG category. V. chlorellavorus contains 13 transposases and 18 genomic islands (genomic regions that are thought to have horizontal origins) (Table 1).

Table 1 Features of the Vampirovibrio chlorellavorus genome.

Isolate name	Vampirovibrio chlorellavorus	
Closest 16S environmental clone a	HG-B02128 (JN409206)	
Number of contigs	26	
Number of plasmids	2	
Total length (bp)	3,030,230	
N50	217,646	
GC (%)	51.4	
tRNA genes	41	
rRNA genes found in genome	16S, 23S	
Putative genes	2,844	
Genomic islands b	18	
Mobile genetic elements	13 transposases	
CDS coding for hydrolytic enzymes	106 proteases/peptidases	
	0 DNases	
	0 RNases	
	0 glycanases	
	3 lipases/esterases	
	2 lysophospholipase	
Genome completeness c	100% (104/104)	
Genome contamination c	0.95% (1/104)	
Proposed class	Melainabacteria	
Proposed order	Vampirovibrionales	
Notes.

a BLASTN search was used to identify the closest neighbour in the May, 2013 version of the Greengenes database (McDonald et al., 2012).

b IslandViewer was used to identify genomic islands (Langille & Brinkman, 2009) with the SIGI-HMM programme (Waack et al., 2006).

c Estimated using CheckM v0.9.5 (Parks et al., 2014a).

Phylogeny and taxonomy

The 16S rRNA gene obtained from the draft genome is identical to the reference sequence for V. chlorellavorus ATCC 29753 (acc. HM038000) and comparative analysis confirmed its placement as a deep-branching member of the Cyanobacteria phylum within the class Melainabacteria and order Vampirovibrionales (Soo et al., 2014; Fig. 1B). Importantly, a concatenated gene tree of 109 conserved single copy genes produced a robust topology consistent with the 16S rRNA tree, also placing V. chlorellavorus in the class Melainabacteria (Fig. 1A: Fig. S2). These phylogenetic inferences clearly indicate that V. chlorellavorus is not a member of the Deltaproteobacteria as first suggested (Gromov & Mamkaeva, 1972).

Figure 1 Phylogenetic position of Vampirovibrio chlorellavorus in the phylum Cyanobacteria.

(A) A maximum likelihood (ML) phylogenetic tree of the phylum Cyanobacteria inferred from a concatenated alignment of 109 single copy marker genes conserved across the bacterial domain. Black circles represent branch nodes with >90% bootstrap support by ML analysis. Class Oxyphotobacteria group names are according to Shih et al. (2013). The blue and red arrow indicate putative acquisition and loss of flagella respectively in the class Melainabacteria. Representatives of 32 bacterial phyla were used as outgroups in the analysis (Fig. S2). Ca, Candidatus. (B) A ML tree of the order Vampirovibrionales (Soo et al., 2014) based on aligned 16S rRNA gene sequences from the May, 2013 Greengenes database (McDonald et al., 2012). Black circles represent nodes with >90% ML, maximum parsimony (MP) and neighbour joining (NJ) bootstrap support values.

Cell shape and envelope

Microscopy studies revealed that V. chlorellavorus has a pleomorphic life cycle, being cocci during its free-living phase and vibrioid once attached to its host (Coder & Starr, 1978). The V. chlorellavorus genome contains genes for the shape-determining protein (mreB) and a key cell division protein (ftsZ), which have been shown to be necessary for the maintenance of cell shape in Caulobacter crescentus and Eschericia coli (Divakaruni et al., 2007; Varma & Young, 2009). The bacterium also contains the genes indicative of a Gram-negative cell envelope including those for the production of lipopolysaccharide (LPS), Lipid A and O-antigen (Beveridge, 1999). This is consistent with prior ultrastructural imaging of V. chlorellavorus which showed this bacterium has a typical Gram-negative cell envelope (Coder & Starr, 1978). Interestingly, the genome also contains surface layer homology (SLH) domains, suggesting that the cell has the capacity to produce an S-layer, although no such structures were observable in transmission electron microscopy (TEM) images (Coder & Starr, 1978; Mamkaeva & Rybal’chenko, 1979). This does not preclude their presence, however, because the samples were not processed optimally for S-layer visualisation; and under unfavourable laboratory cultivation conditions, the formation of the S-layer may be lost (Sára & Sleytr, 2000; Šmarda et al., 2002). S-layers have been observed in at least 60 strains of Cyanobacteria (Šmarda et al., 2002) and SLH domains have also been found in other Melainabacterial genomes.

Core metabolism

The V. chlorellavorus genome encodes a complete glycolysis pathway utilising glucose-6-phosphate, glycerol and mannose, the pentose phosphate pathway and a tricarboxylic acid (TCA) cycle. The genome also contains a complete set of genes for an electron transport chain comprising Complexes I to IV and an F-type ATPase. It has two terminal oxidases; a bd-type quinol and a cbb3-type cytochrome (Complex IV), both of which are used for microaerobic respiration (Preisig et al., 1996). According to PHX (predicted highly expressed) analysis (Karlin & Mrázek, 2000), many of the genes in the glycolysis pathway, TCA cycle and electron transport chain are predicted to be highly expressed (Fig. 2; Table S2) suggesting oxidative metabolism is central to the predatory lifestyle of V. chlorellavorus despite the inference of adaptation to low oxygen conditions. However, the genome also contains lactate dehydrogenase suggesting that it is able to ferment pyruvate to lactate under anaerobic conditions (Fig. 2). The bacterium contains genes for fatty acid biosynthesis and β-oxidation, which leads to the production of acetyl-CoA. Consistent with other described members of the class Melainabacteria, and in contrast to oxygenic photosynthetic cyanobacteria, V. chlorellavorus lacks genes for photosynthesis and carbon fixation (Soo et al., 2014). V. chlorellavorus can synthesise its own nucleotides and several cofactors and vitamins including lipoate, nicotinate, heme, riboflavin and thiamine-diphosphate, but only 15 amino acids: alanine, asparagine, aspartate, cysteine, glutamate, glutamine, glycine, isoleucine, leucine, lysine, methionine, proline, threonine, tryptophan and valine. Although V. chlorellavorus does not have the genes necessary to synthesise the remaining five amino acids or their polyamine derivatives, it contains amino acid and polyamine transporters (Fig. 2) that would allow it to obtain these organic compounds from external sources, most likely C. vulgaris.

Figure 2 Metabolic reconstruction of Vampirovibrio chlorellavorus.

Metabolic predictions for V. chlorellavorus based on genes annotated by IMG/ER (Markowitz et al., 2009). Solid and dashed lines represent single or multiple steps in a pathway respectively. Black ovals indicate substrates that enter the glycolysis pathway. Fermentation end-products are indicated as black rectangles. V. chlorellavorus is capable of oxidative phosphorylation as it contains a complete TCA cycle and electron transport chain. Biosynthetic products are shown in green (amino acids), red (co-factors and vitamins), purple (nucleotides), and orange (non-mevalonate pathway products). Serine (highlighted in blue) is not able to be synthesised and is presumably transported into the cell. ATP-binding cassette transporters are highlighted in yellow and permeases, pumps and transporters are highlighted in orange. The direction of substrate transport across the membrane is shown with arrows. Putatively highly expressed genes and complexes are bolded. V. chlorellavorus is missing all recognised photosynthesis genes including those for Photosystems I and II, chlorophyll and antennae proteins.

The predatory lifestyle of Vampirovibrio chlorellavorus

Based on genomic inference and electron microscopy images obtained by Coder & Starr (1978), we divide the predatory life cycle of V. chlorellavorus into five phases comprising (i) prey location, (ii) attachment and formation of secretion apparatus, (iii) ingestion, (iv) binary division and (v) release (Fig. 3).

Figure 3 Proposed predatory life cycle of Vampirovibrio chlorellavorus informed by genome annotations.

(i) V. chlorellavorus seeks out C. vulgaris cells via chemotaxis and flagella. (ii) It attaches to prey cells via a type IV secretion system (T4SS). (iii) Plasmid DNA and hydrolytic enzymes are transferred to the prey cells via the T4SS where they degrade algal cell contents (see Fig. 4 for details). (iv) Algal cell exudates are ingested by V. chlorellavorus allowing it to replicate by binary division. (v) Progeny are released completing the cycle. S, starch granule; M, mitochondria; N, nucleus.

Phase i: prey location

The V. chlorellavorus genome encodes two-component regulatory systems including the well-known CheA-CheY signal transduction pathway that couples to flagella rotation or pili extension, attachment and retention (Fig. 2) allowing the cell to move towards chemoattractants or away from chemorepellents (Wadhams & Armitage, 2004). Coder & Starr (1978) showed that V. chlorellavorus is able to swim towards its prey using a single, polar unsheathed flagellum possibly assisted by pili visible as thick bundles in proximity to the flagellum. All of the genes necessary to produce a functional flagellum and type IV pili (TFP) are present in the V. chlorellavorus genome (Macnab, 2003; Table S3). In Cyanobacteria, Synechocystis strain PCC 6803 uses TFP for motility and it has also been speculated that TFP can drive motility in Nostoc punctiforme (Bhaya et al., 2000; Duggan, Gottardello & Adams, 2007). It is likely that V. chlorellavorus uses chemotaxis to help it locate prey, but based on genome inference alone, it is not possible to determine which gradients V. chlorellavorus is detecting and responding to. However, the genome does contain one globin-coupled sensor inferred to be used for aerotaxis (Freitas, Hou & Alam, 2003; Fig. S3) and one putative light-activated kinase (bacteriophytochrome; Bhoo et al., 2001; BphP in Fig. 2) that may enable V. chlorellavorus to move towards oxic and illuminated regions of its habitat that have a higher likelihood of containing Chlorella cells.

Phase ii: attachment and formation of a conjugative secretion apparatus

V. chlorellavorus has a number of cellular features that likely facilitate its observed attachment to Chlorella cells: TFP (described above), an outer membrane protein (OmpA) and von Willebrand domain-containing proteins. While there are no reports of bacteria adhering to unicellular microbial eukaryotes using these structures, there are a number of examples for adherence to animal tissues. TFP are known to be involved in adhesion of pathogenic Escherichia coli and Neisseria meningitidis to human epithelial cells as a key virulence mechanism (Chamot-Rooke et al., 2011; Pizarro-Cerdá & Cossart, 2006). OmpA porins are outer membrane proteins that assemble into an eight stranded β-barrel structure with four surface-exposed loops. Shin et al. (2005), showed that OmpA surface loops are critical for adhesion of E. coli to brain microvascular endothelial cells leading to neonatal meningitis (Shin et al., 2005). Furthermore, OmpA is involved in the binding of Acinetobacter baumanii and Pasteurella multocida to fibronectin from human lung carcinoma (Smani, McConnell & Pachón, 2012). The von Willebrand factor A (VWA) domains are found predominantly in cell adhesion and extracellular matrix molecules, including integrins, hemicentins and matrilins (Whittaker & Hynes, 2002). Enterococcus faecalis VWA domains are able to mediate protein–protein adhesion through a metal ion-dependent adhesion site (Nielsen et al., 2012).

Ultrastructural studies have shown that V. chlorellavorus forms a discrete pad of unknown composition during attachment to Chlorella cells (Gromov & Mamkaeva, 1972; Mamkaeva & Rybal’chenko, 1979). Similar pads are involved in the attachment of the uncultured predatory bacterium Vampirococcus to its bacterial prey, Chromatium (Guerrero et al., 1986). Spikes of electron dense material have been observed to extend from the V. chlorellavorus pad into the Chlorella cell through the algal cell envelope (Coder & Starr, 1978). We propose that the attachment pad and spike are a type IV secretion system (T4SS) fully encoded in the V. chlorellavorus genome in three operons (Fig. 2 and Fig. S4). Phylogenetic analysis of the VirB4 ATPase (gene trbE), a highly conserved component of the T4SS used to classify these secretion systems (Guglielmini, de la Cruz & Rocha, 2013) showed that the V. chlorellavorus orthologue is most closely related to a T-type conjugation system in Nitrosomonas eutropha (Fig. S5). T-type conjugation T4SS are best known in Agrobacterium tumefaciens which form a secretion channel through which the T-strand (the strand destined for transfer) is passed into plant cells causing crown gall disease (Christie, 2004). More generally, T-type conjugation systems can pass single stranded DNA and proteins into recipient cells (Alvarez-Martinez & Christie, 2009). Two of the T4SS operons of V. chlorellavorus are found on conjugative plasmids (Fig. S4), which are predicted to be made singlestranded by their relaxases, nicking the DNA at the origin of transfer and transporting the T-strand to the Chlorella cell via the conjugation channel. The T-strand would then integrate into the Chlorella chromosome and be expressed (Cascales & Christie, 2003) (Fig. 4). Since the nature of the relationship between the two conjugating cells is predatory, we may expect that the T-strand would carry genes that facilitate ingestion of the Chlorella cell contents. No genes encoding hydrolytic enzymes were identified on the plasmids, though one encodes several efflux transporters (Fig. S4).

Figure 4 Proposed conjugative mechanism.

T4SS operons are found on two conjugative plasmids in V. chlorellavorus. The T-strands of the plasmids are predicted to be made single-stranded by plasmid-encoded relaxases (Fig. S4), nicking the DNA at the origin of transfer and transporting the T-strands to the C. vulgaris cell via the mating pair formation. We predict that effector proteins (hydrolytic enzymes) synthesised in the bacterium are also transported via the mating pair formation. The T-strand enters the algal nucleus through a nuclear pore complex and is incorporated into a C. vulgaris chromosome. The effector proteins degrade the algae contents which are transported out of the algal cell via T-strand encoded transporters (Fig. S4). The algal lysates are imported into the V. chlorellavorus cell providing energy and nutrients for replication.

Phase iii: ingestion

Five to seven days after V. chlorellavorus attachment, Chlorella cells remain intact but are devoid of cytoplasmic contents and contain only large vacuolated areas and membranous structures which are presumed to be organellar remains (Coder & Starr, 1978). The V. chlorellavorus genome encodes numerous proteins that may be involved in the observed ingestion of Chlorella cell contents, including 108 proteases and 123 carbohydrate-active enzymes (Tables S4 and S5). The majority of the latter group are glycoside hydrolases which are predicted to degrade polysaccharides and glycoproteins, major components of the Chlorella cell envelope (Gerken, Donohoe & Knoshaug, 2013) as well as starch and glycogen, which are diurnally stored as energy sources in Chlorella (Nakamura & Miyachi, 1982). Extracellular proteases are produced by many bacterial pathogens and are commonly involved in the degradation of the host extracellular matrix, facilitating invasion and colonisation (Kennan et al., 2010). They have also been suggested as important factors in virulence for other predatory bacteria, for example Bdellovibrio bacteriovorus and Micavibrio aeruginosa (Rendulic et al., 2004; Wang, Kadouri & Wu, 2011). The V. chlorellavorus genome contains an alginate lyase, an enzyme that is able to degrade alginate via β-elimination cleavage of glycosidic bonds in the polysaccharide backbone (Lamppa et al., 2011). Alginate is a common component of marine brown algae cell envelopes and intracellular material which is targeted as a carbon and energy source by bacteria possessing alginate lyases (Wong, Preston & Schiller, 2000). Chlorella cells may similarly contain alginate supported by the finding of an alginate lyase gene in a Chlorella virus (Suda et al., 1999). We propose that this suite of hydrolytic enzymes are synthesised in V. chlorellavorus and transported via the T4SS conjugation channel into the prey cell where they produce hydrolysates in the Chlorella cell (Fig. 3). The T4SS plasmid-encoded efflux transporters (Fig. S4) may facilitate the export of lysates from the Chlorella cell assuming that the T-strand is integrated and expressed in Chlorella as is the case in Agrobacterium tumour formation (Christie, 2004). Lysates exported into the surrounding milieu could then be imported into the attached V. chlorellavorus cell (and possibly neighbouring predatory cells) using a number of transport systems from the ATP-binding cassette (ABC) superfamily, the Major Facilitator Superfamily and/or permeases encoded in the bacterial genome (Fig. 2). It is unlikely that Chlorella lysates would be directly transported into V. chlorellavorus cells via the conjugation channel as conjugation systems have only been shown to deliver protein or DNA substrates to eukaryotic target cells but not vice versa (Cascales & Christie, 2003).

Phase iv: binary fission

Attached V. chlorellavorus cells have been observed to divide by binary fission presumably using nutrients and energy derived from ingestion of Chlorella lysates, consistent with an obligate predatory lifestyle (Coder & Starr, 1978; Gromov & Mamkaeva, 1980). The genome contains the cell division proteins required to replicate by this process, including the tubulin-like protein FtsZ, which is predicted to be highly expressed by PHX analysis, and the regulation of the placement of division site genes, minC, -D and -E (Lutkenhaus & Addinall, 1997).

Phase v: release

A new lifecycle is started when progeny cells release from consumed Chlorella cells (Fig. 3). Released cells then synthesise flagella to aid their dispersal and have a range of mechanisms to protect themselves from environmental stress as free-living organisms. The V. chlorellavorus genome encodes two superoxide dismutases, which convert O2− to H2O2 and O2 (Cabiscol, Tamarit & Ros, 2000) and one catalase-peroxidase, katG, a H2O2 scavenger (Jittawuttipoka et al., 2009). Both of these enzymes can be used to combat oxidative stress that may be induced by environmental agents such as radiation or compounds that can generate intracellular O2− (Cabiscol, Tamarit & Ros, 2000) or from the Chlorella (Mallick & Mohn, 2000). The genome encodes a large and small conductance mechanosensitive channel protein that prevents cells from lysing upon sudden hypo-osmotic shock by releasing solutes and water (Birkner, Poolman & Koçer, 2012). It also encodes a protein containing a stress-induced bacterial acidophilic repeat motif and three copies of a universal stress protein (UspA), an autophosphorylating serine and threonine phosphoprotein (Kvint et al., 2003). In other stress conditions, such as temperature shock, starvation or the presence of oxidants or DNA-damaging agents, the expression of UspA is increased or decreased, which is known to be correlated with improved bacterial survival (Jenkins, Burton & Cooper, 2011). Beta-lactamases, cation/multidrug efflux pumps and ABC-type multidrug and solvent transport systems were identified (Fig. 2) that could be used to eliminate antibiotics or toxins encountered in the environment (Frère, 1995; Lubelski, Konings & Driessen, 2007).

Comparison of V. chlorellavorus to other predatory bacteria

A study of 11 predatory and 19 non-predatory bacterial genomes was conducted to define the ‘predatome’, the core gene set proposed for bacteria with predatory lifestyles (Pasternak et al., 2013). The study found that the most striking difference between predators and non-predators is their method of synthesising isoprenoids. All predators, except for M. aeruginosavorus, encode the three essential enzymes used in the mevalonate pathway, which is uncommon in bacteria, whereas non-predators encode five essential enzymes for the more typical non-mevalonate pathway. It was suggested that predatory bacteria may have access to acetoacetyl-CoA pools in their prey cells, which is the first substrate used in the mevalonate pathway (Pasternak et al., 2013). However, V. chlorellavorus lacks two of the three mevalonate pathway genes and instead encodes the non-mevalonate pathway (Fig. 2). Twelve additional protein families were identified as specific to the predator set including those involved in chemotaxis, cell adhesion, degradation of polypeptides and benzoate, and four enzymes that may have evolved to scavenge essential metabolites (Pasternak et al., 2013). V. chlorellavorus has orthologues of eight of these protein families, and while lacking some of the specific adhesion and degradation genes (OrthoMCL OG4 39191, 26993, 21243, 18254), it encodes alternative proteins for these functions (see above). Eleven additional protein families were identified as specific to the non-predatory bacteria including those for riboflavin and amino acid synthesis, specifically tryptophan, phenylalanine, tyrosine, valine, leucine and isoleucine (Pasternak et al., 2013). V. chlorellavorus has all but one of these “non-predatory” genes (OrthoMCL OG4 11203) which may reflect its phylogenetic novelty given that the core set analysis was based mostly on comparison of Proteobacteria (Pasternak et al., 2013). We note that while V. chlorellavorus can make these particular compounds, its cofactor and amino acid biosynthesis repertoire is limited (5 cofactors, 15 amino acids).

Comparison of V. chlorellavorus to other Melainabacteria genomes

Consistent with all sequenced representatives of the class Melainabacteria (Di Rienzi et al., 2013; Soo et al., 2014), V. chlorellavorus is missing all recognised photosynthesis genes including those for Photosystems I and II, chlorophyll and antennae proteins. This supports the hypothesis that photosynthetic cyanobacteria acquired photosystems after diverging from the ancestor of the Melainabacteria (Di Rienzi et al., 2013; Soo et al., 2014; Fig. 1). The V. chlorellavorus genome falls within the size range of previously reported Melainabacteria (1.8 to 5.5 Mbp) but has the highest GC content thus far (51.4%) compared with the GC content of other Melainabacteria who have a range of 27.5% to 49.4%. V. chlorellavorus is the second representative of the class inferred to be capable of oxidative phosphorylation as it contains a full respiratory chain (Fig. 2), the other being Obscuribacter phosphatis (Soo et al., 2014). V. chlorellavorus encodes a flagellum which is also found in some representatives of the order Gastranaerophilales (ACD20, MEL_B1 and MEL_B2). We inferred a phylogenetic tree for the conserved flagella marker gene, fliI (Minamino & Namba, 2008) and found that the Melainabacteria fliI genes form a monophyletic cluster consistent with their internal branching order in the genome tree (Fig. 1 and Fig. S6) This association suggests that flagella were present in the cyanobacterial ancestor of the Gastranaerophilales and Vampirovibrionales and were subsequently lost at least once in the Gastranaerophilales (Fig. 1). A global comparison of COG (clusters of orthologous groups) categories revealed that V. chlorellavorus has a functional distribution typical of other Melainabacteria genomes with the exception of genes involved in intracellular trafficking, secretion, and vesicular transport (Fig. S7). V. chlorellavorus is overrepresented in this category due to a higher proportion of genes involved in Type IV secretion systems, which we posit to be important in the lifecycle of this predator (see above).

Conclusions

We have sequenced and assembled a near complete genome from a 36-year old lyophilised co-culture of the predatory bacterium Vampirovibrio chlorellavorus. Comparative gene and genome analyses confirm that V. chlorellavorus is a member of the Melainabacteria, a recently described non-photosynthetic class in the cyanobacterial phylum (Soo et al., 2014). V. chlorellavorus is the first recognised member of the Cyanobacteria with a predatory lifecycle and we predict that it is the first predator to use a conjugative type IV secretion system similar to Agrobacterium tumefaciens to invade its host. It remains to be determined how widespread this phenotype is within the Melainabacteria and how it may have evolved from non-predatory cyanobacterial ancestors.

Supplemental Information

Figure S1 Depth of coverage against GC content for the assembled contigs

Contigs assigned to V. chlorellavorus are represented by red circles. C. vulgaris chloroplast contigs are represented by green circles and C. vulgaris contigs are represented by black circles. The size of the circle corresponds to the length of the contig.

Click here for additional data file.

Figure S2 A maximum likelihood concatenated gene tree showing the Cyanobacteria and select bacterial phyla

The phylogenetic tree was inferred from the concatenation of 109 conserved marker genes (Table S1) and consists of 7,732 bacterial and 169 arachael genomes from the IMG database (Markowitz et al., 2014).

Click here for additional data file.

Figure S3 Methyl-accepting chemotaxis proteins (MCP) encoded in the V. chlorellavorus genome

Three MCPs are encoded in the V. chlorellavorus genome. MCP domains were predicted with InterProScan5 (Jones et al., 2014). Directional arrows represent the putative genes and orientation (positive or negative strand) and the rounded rectangles represent domains.

Click here for additional data file.

Figure S4 Type IV secretion system (T4SS) in the V. chlorellavorus genome and plasmids

The schematic diagram shows the presence of T4SS genes identified by IMG/ER on both plasmids and one contig. The arrows represent annotated genes and their direction. The numbers above the genes are the IMG/ER accession numbers and these have been identified as alien genes by PHX analysis (Table S2). T4SS genes have the predicted gene names italicised below.

Click here for additional data file.

Figure S5 A maximum likelihood phylogenetic tree of virB4 genes

Aligned sequences and naming conventions were obtained from Christie (2004). virB4_T is based on the T-DNA conjugation system of Agrobacterium tumefaciens plasmid Ti, virB4_F is based on the plasmid F, virB4_I is based on the Incl plasmid R64 and virB4_G is based on ICEHIN1056. The other T4SS have homologues to VirB4 and include the Cyanobacteria (virB4_C), Bacteroides (virB4_B), Firmicutes (virB4_FA and virB4_FATA), Actinobacteria (virB4_FA and virB4_FATA), Tenericutes (virB4_FATA) and Archaea (virB4_FATA) (Christie, 2004). The V. chlorellavorus genome contains T4SS that belong to the virB4_T. virD4 is used as the outgroup. Black circles represent nodes with ≥90% bootstrap support, grey circles represent nodes with ≥ 80% bootstrap support and white circles represent nodes with ≥70% bootstrap support. (p) corresponds to virB4 genes found on plasmids and (c) corresponds to virB4 genes found on the chromosome.

Click here for additional data file.

Figure S6 A maximum likelihood phylogenetic tree of fliI genes

The phylogenetic tree is constructed from 2,256 finished genomes from the IMG database (Markowitz et al., 2009). The tree is unrooted and only the Melainabacteria and its closest neighbours are shown. V. chlorellavorus is in red and the other three Melainabacteria representatives are in blue. Phyla are in bold. Black circles in the tree represents nodes with ≥90% bootstrap support and white circles represents nodes with ≥80% bootstrap support.

Click here for additional data file.

Figure S7 Clusters of orthologous groups (COGs) for the class Melainabacteria

There is an increase in the COGs associated with cell motility (category N) and intracellular trafficking, secretion, and vesicular transport (category U) for V. chlorellavorus. STAMP (Parks et al., 2014a; Parks et al., 2014b) was used to explore the resulting COG profiles and create summary plots.

Click here for additional data file.

Table S1 Marker genes used for constructing the concatenated gene tree

A set of 178 single copy genes present exactly once in >90% of the trusted genomes (found in >90% of the genomes) from the Integrated Microbial Genomes (IMG; Markowitz et al., 2014) database was identified. From the 178 initial genes, 69 were removed from consideration as they exhibited divergent phylogenetic histories in >1% of the trusted genomes. The remaining 109 genes were used to construct a concatenated gene tree (Fig. S2).

Click here for additional data file.

Table S2 Predicted highly expressed (PHX) and alien genes

PHX and alien gene prediction was performed with PHX analysis, using ribosomal proteins, chaperones and transcriptional and translational proteins of V. chlorellavous as representatives of recognised highly expressed genes to identify other putatively highly expressed genes in the genome (Karlin & Mrázek, 2000).

Click here for additional data file.

Table S3 Flagella and type IV pili genes encoded in the V. chlorellavorus genome

Putative gene numbers are assigned using IMG/ER (Markowitz et al., 2009).

Click here for additional data file.

Table S4 Carbohydrate-active enzymes and proteases encoded in the V. chlorellavous genome

Putative genes were annotated with the dbCAN web server (Yin et al., 2012) to identify glycoside hydrolases and checked against the IMG annotations and BLAST results.

Click here for additional data file.

Table S5 Peptidases encoded in the V. chlorellavorus genome

The MEROPS server (Rawlings et al., 2014) was used to identify putative peptidases in V. chlorellavorus using batch BLAST.

Click here for additional data file.

We thank Jim Prosser, Samantha Law and Tina Niven from NCIMB for their help with obtaining the co-cultures of V. chlorellavorus and C. vulgaris and Serene Lowe for preparing the DNA for sequencing and IMB, UQ for sequencing. We also thank Xuyen Le and Bryan Wee for discussions on motility and T4SS, Julien Guglielmini for data on T4SS, Rick Webb for inspection of S-layers in transmission electron microscopy images, Michael Nefedov for translation of Russian manuscripts and Nancy Lachner for attempts to extract RNA from the lyophilised cells.

Additional Information and Declarations

Competing Interests

Author Contributions

DNA Deposition

The authors declare there are no competing interests.

Rochelle M. Soo conceived and designed the experiments, performed the experiments, analyzed the data, wrote the paper, prepared figures and/or tables, reviewed drafts of the paper.

Ben J. Woodcroft performed the experiments, analyzed the data, contributed reagents/materials/analysis tools, reviewed drafts of the paper.

Donovan H. Parks performed the experiments, analyzed the data, contributed reagents/materials/analysis tools, prepared figures and/or tables, reviewed drafts of the paper.

Gene W. Tyson conceived and designed the experiments, contributed reagents/materials/analysis tools, reviewed drafts of the paper.

Philip Hugenholtz conceived and designed the experiments, contributed reagents/materials/analysis tools, wrote the paper, reviewed drafts of the paper.

The following information was supplied regarding the deposition of DNA sequences:

Database: JGI IMG ER

Accession number: 2600254900

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
