# Peer review of "Back from the dead; the curious tale of the predatory cyanobacterium Vampirovibrio chlorellavorus"

_PeerJ, doi:10.7717/peerj.968_

## Round 0.1 · original submission · Minor Revisions

This paper has been well-received. However, minor revisions are necessary. I think the improvements suggested by the reviewers, while clarifying the technical details, will help to further highlight the importance of this work.

Reviewer 1 ·

Basic reporting

The paper was a pleasure to read, being well written and with sufficient background information and context of the work to produce a really interesting manuscript. The authors present a genome of the novel cyanobacterial class Melainabacteria which they obtained following DNA extraction of lyophilised material from an old co-culture with its Chlorella host. Using this genomic information the authors perform detailed analysis in the context of what is known of the V. chlorellavorus lifestyle.

Experimental design

The methods are sufficiently described.

Validity of the findings

The manuscript is mostly very well supported by the information presented. One query though is about genome coverage and closure, and the subsequent case made for missing genes. So, I fully take the basis on which the authors base their genome assembly (coverage, GC% etc) but I am less clear about the size of any gaps (lines 96-99 and lines 221-225). This has implications for those genes that are stated as missing. Related to this it was unclear why no 5S rRNA gene was identified.

I was also interested by the mention (in the acknowledgements) that RNA extraction was attempted from the lyophilised sample. Did this work? & if so, have the authors any data to show that any of the many interesting genes they describe (e.g. the type IV TSS) are expressed?

Additional comments

Minor comments:

Line 46: …in 1978. However,….
Line 59: (Soo et al., 2014), ……i.e. comma not semi-colon after brackets
Line 60, Here, we report….
Line 91: Additional file 1 should be replaced by Supplementary Figure 1
Lin 103: Woodcroft et al., unpublished (not in prep)
Line 113: …was submitted to IMG/ER….
Line 135: replace blasted with were subjected to BLAST analysis….
Line 170: alignments
Line 182: Guglielmini et al., (2012) i.e. remove repetition
Line 254: ….which showed this bacterium has a ….
Line 310: give year of the Freitas et al reference (same in reference list)
Line 317: cells: TFP…. (i.e. colon not semi-colon)
Line 334: forms a …
Line 416: Chlorella (in italics)
Line 426: Fig 2 not Fig 3
Line 551: R. Kolter doesn’t appear to be an author on the Di Rienzi et al., paper so should be removed
Line 716: give volume and page numbers for this reference
Line 723: Chlorella in italics

·

Basic reporting

This is a a very well written manuscript that provides sufficient background information to frame the scientific story and make it interesting. No major changes in structure or content are necessary.

Experimental design

The bioinformatics approaches used are appropriate and well executed.

Validity of the findings

The findings are interesting and described in a lucid manner that is both explanatory and thought provoking.

Additional comments

The manuscript by Soo and colleagues describes a most interesting story of DNA detective work resulting in the reconstruction of a predatory non-photosynthetic cyanobacterial genome from a lyophilized stock culture. The genome of Vampirovibirio chlorellavorus reveals specific metabolic adaptations to this predatory lifecycle including the use of a type IV secretion system reminiscent of the T-plasmid used by Agrobacterium. Overall this is a very well written manuscript that clearly reconstructs the core metabolic properties and putative predatory process of a basal cyanobacterium using appropriate bioinformatics approaches, lucid graphics and a creative edginess that engages the reader throughout. It will be interesting to see if the authors are able to identify living co-culture systems in the future to directly test some of their hypotheses.

Some minor editorial changes are suggested to improve the flow of ideas.

Page 3 Lines 45-50. The transition from successful resuscitation to subsequent studies with co-cultures to the ATCC sequencing the 16S rRNA gene is choppy. What happened to prevent future work with co-cultures? Why did cultivation-independent methods become a prerequisite to study this organism?

Page 3 Line 60. In the methods the authors refer to the genome as a population bin but in the main text they refer to a “complete genome”. It would be good to pick a single use term e.g. genome or population genome and refer to this term throughout the manuscript for consistencies sake. Was there any evidence for population heterogeneity in the assembled contigs?

Page 6 Line 134. Please explain what is meant by “highly expressed and alien genes”. Explain how this analysis of ribosomal proteins, chaperones and information processing genes connect to alien genes?

Page 7 Line 151. Please indicate the 178 initial and subsequent 69 genes that were removed from consideration.

Page 10 Line 222. Here you refer to a draft genome. Please refer to comment for Page 3 Line 60. Pick a consistent use term for the construct and use throughout the manuscript.

Page 12 Line 284. Of the “five remaining amino acid” biosynthetic pathways did you identify genes encoding any reactions within these pathways suggesting interpathway complementation with the host or other microorganisms in the co-culture?

Page 13 Line 303. This sentence reads somewhat circular. Can you be more specific as to the criteria used to make this prediction?

Page 18 Line 412. The O2- superscript is really hard to discern for superoxide.

Page 18 Line 412. The katG gene should be in italics

Page 18 Line 416. “The bacterium…” harbors genes encoding the proteins indicated. Not sure that you can indicate that it contains specific proteins at this point.

Page 18 Line 424. Here “the bacterieum encodes…” Be consistent with the way in which you refer to the genomic potential of this microorganism.

Figure 1. What is the red arrow pointing at? For the inset why not indicate the V. chlorellavorus branch as a red triangle that you then open up in the inset?

---

## Round 0.2 · accepted · Accept

I feel that the modified manuscript makes a great contribution to PeerJ and to the discipline.